# The Striking Flower-in-Flower Phenotype of *Arabidopsis thaliana* Nossen (No-0) is Caused by a Novel *LEAFY* Allele

**DOI:** 10.3390/plants8120599

**Published:** 2019-12-13

**Authors:** Anne Mohrholz, Hequan Sun, Nina Glöckner, Sabine Hummel, Üner Kolukisaoglu, Korbinian Schneeberger, Klaus Harter

**Affiliations:** 1Center for Plant Molecular Biology, Eberhard Karls University Tübingen, 72076 Tübingen, Germanynina.gloeckner@zmbp.uni-tuebingen.de (N.G.); sabine.hummel@zmbp.uni-tuebingen.de (S.H.); uener.kolukisaoglu@zmbp.uni-tuebingen.de (Ü.K.); 2Department of Plant Developmental Biology, Max Planck Institute for Plant Breeding Research, 50829 Cologne, Germany; sun@mpipz.mpg.de (H.S.); schneeberger@mpipz.mpg.de (K.S.)

**Keywords:** *Arabidopsis thaliana*, floral development, flower morphology, *Ds* transposon, classical/sequencing-based mapping, *LEAFY*, DNA-binding

## Abstract

The transition to reproduction is a crucial step in the life cycle of any organism. In *Arabidopsis thaliana* the establishment of reproductive growth can be divided into two phases: Firstly, cauline leaves with axillary meristems are formed and internode elongation begins. Secondly, lateral meristems develop into flowers with defined organs. Floral shoots are usually determinate and suppress the development of lateral shoots. Here, we describe a transposon insertion mutant in the Nossen accession with defects in floral development and growth. Most strikingly is the outgrowth of stems from the axillary bracts of the primary flower carrying secondary flowers. Therefore, we named this mutant flower-in-flower (*fif*). However, the transposon insertion in the annotated gene is not the cause for the *fif* phenotype. By means of classical and genome sequencing-based mapping, the mutation responsible for the *fif* phenotype was found to be in the *LEAFY* gene. The mutation, a G-to-A exchange in the second exon of *LEAFY*, creates a novel *lfy* allele and results in a cysteine-to-tyrosine exchange in the α1-helix of LEAFY’s DNA-binding domain. This exchange abolishes target DNA-binding, whereas subcellular localization and homomerization are not affected. To explain the strong fif phenotype against these molecular findings, several hypotheses are discussed.

## 1. Introduction

The development of flowers is indispensable for the reproductive success of angiosperm plants. During vegetative growth, the shoot apical meristem (SAM) develops leaves and/or branches, the latter with their own SAMs. After the switch to reproductive growth, the apical meristems give rise to flowers. Floral development differs crucially from vegetative shoot growth, as the flower possesses several types of organs of which the number, arrangement and morphology are species-specific. Furthermore, the development of lateral shoots is inhibited in flowers and floral shoots are determinate after the last reproductive organs have been initiated [1,2,3]. Thus, the coordination of complex molecular processes is necessary for successful floral development.

There has been significant progress in recent years towards understanding the molecular mechanisms underlying flower formation [4,5,6,7,8]. Central to this was the identification and cloning of the genes that initiate and maintain floral development in plant species, including *Arabidopsis thaliana*. The most intriguing discovery was the *Arabidopsis* loss-of-function mutants with structures that are intermediate between floral and vegetative shoots. The cloning of the corresponding genes revealed the existence of the master regulators required for the floral initiation process. To date, five regulatory master genes are known: *LEAFY* (*LFY*), *APETALA1* (*AP1*), *CAULIFLOWER* (*CAL*), *APETALA2* (*AP2*) and *UNUSUAL FLORAL ORGANS* (*UFO*) [3]. *LFY* and *AP1* play a primary role in initiating the floral program, as the corresponding loss-of-function mutants do not generate shoots with floral characteristics and the ectopic expression of either gene induces precocious flower formation [9,10,11]. *LFY*, *AP1* and *CAL* encode for transcription factors and are expressed predominantly in floral primordia [12,13,14].

During plant vegetative growth, *LFY* expression increases in newly formed leaves until a certain threshold is reached [15]. LFY then induces the expression of *AP1*/*CAL* genes by activation of the *AP1*/*CAL* promoters. Through their mutual transcriptional up-regulation, LFY and AP1/CAL cooperate to cause the floral transition [16,17]. Once the floral meristem is established, the floral initiation gene functions govern its spatial patterning by inducing the expression of the floral homeotic *ABC* genes, such as *AP2*, *AP3*, *Pistillata* (*PI*) and *AGAMOUS* (*AG*). The *ABC* gene functions in turn control the identity of the stereotypically arranged *Arabidopsis* floral organs [18,19].

In the course of our study of the influence of abiotic stress on flower symmetry, we searched for novel insertion mutants with defects in floral development or morphology in different *Arabidopsis thaliana* accessions. We focused on genes that had not yet been linked to flowering. A *Ds* transposon insertion mutant, which developed secondary inflorescences with partially aberrant flowers and had an aberrant growth phenotype, was identified in the No-0 accession. The wild type allele of the gene carrying the *Ds* transposon codes for a cystein/histidine-rich C1 domain protein [20,21]. However, a thorough genetic analysis revealed that the transposon-inserted allele is not the cause of the observed floral and developmental phenotype. Using classical mapping and mapping-by-sequencing, we eventually found a novel mutant allele of *LFY* to be responsible for the aberrant floral development, flower morphology and addressed the molecular reason for LFY malfunction.

## 2. Results

### 2.1. The Flower-in-Flower (fif) Transposon Insertion Line Displays a Novel Flower Phenotype

In order to identify novel *Arabidopsis thaliana* mutants with defects in flowering we screened the RIKEN Arabidopsis Phenome Information Database (RAPID; [22]). RAPID also covers a Ds transposon mutant collection in the *Arabidopsis* Nossen-0 (No-0) background [23,24]. We identified a transposon-tagged line (15-3794-1), which developed secondary inflorescences with partially aberrant flowers (Figure 1a). Because of this phenotype, we named this *Arabidopsis mutant flower-in-flower* (*fif*).

Wild type *Arabidopsis* flowers does not have bracts but consist of four concentric rings of four sepals, four petals, six stamens and two fused carpels (Figure 1c). In contrast, the primary flower of the *fif* mutant had bracts as well as sepals but the petals were incompletely developed or entirely missing (Figure 1b). In addition, there were either no stamens or the stamens displaying an aberrant development (Figure 1b). Furthermore, there were more than two carpels per flower, which were not or only partially overgrown and sterile. Most obvious, however, was the outgrowth of stems from the axillary meristems of the leaf-like sepal structures, which carried terminal secondary flowers (1d). A few secondary *fif* flowers showed a wild type-like phenotype and were fertile (Figure 1b,d).

Furthermore, the *fif* mutant plant displayed an aberrant growth habitus compared to wild type No-0 (Figure 2a,b). The “bushy” appearance of the *fif* mutant was due to an enhanced number of stem-born side branches compared to wild type No-0, whereas the number of rosette-born side shoots was the same in *fif* and wild type No-0 plants (Figure 2c). Furthermore, the stem length of the *fif* mutant plants a month after sowing showed a different size distribution compared to wild type No-0 and Col-0 (Figure 2d). Whereas the majority of the stem length of the No-0 and Col-0 plants was at 0 cm and between 0.5 and 3.0 cm, the *fif* plants by majority exhibited stem lengths between 3 and 8 cm (Figure 2d). In addition, the *fif* mutant plants showed extended vegetative growth and, therefore, delayed flowering compared to wild type No-0 and Col-0 as indicated by the enhanced number of rosette leaves until flowering (Figure 2e).

### 2.2. The Transposon Insertion is Not Responsible for the fif Phenotype

According to the RIKEN RAPID and our own genotyping results, the *Ds* transposon was located in the second exon of the gene *At1g20990* that codes for a putative cysteine/histidine-rich C1 domain protein with a yet unknown function. To validate, whether the mutant allele of the *At1g20990* locus is actually the cause for the *fif* phenotype, we performed a (co-) segregation analysis by backcrossing the *fif* mutant with wild type No-0 in both directions (♀*fif* × ♂No-0, ♀No-0 × ♂*fif*). Irrespective of the direction, the crosses were successful as demonstrated by PCR on genomic DNA extracted from F1 plants using *Ds* transposon- and *At1g20990*-specific primers (Appendix A). All tested F1 plants were heterozygous for the *Ds* transposon and wild type *At1g20990* and displayed wild type floral organs and growth (Appendix A). Therefore, the mutation that causes the *fif* phenotype is recessive. Next, six F1 plants were self-fertilized and 20 to 30 progenies each analysed for their pheno- and genotypes, respectively. As shown in Figure 3a, around one quarter of the F2 plants displayed the *fif* phenotype indicating that it is caused by a single mutant gene. Intriguingly, our genotyping results showed that the *Ds* transposon insertion did not co-segregate with the *fif* phenotype: 29% of the *fif* phenotype-displaying plants did not contain the transposon, an additional 45.8% contained the transposon insertion only heterozygously (Figure 3a). These results prove that the *Ds* insertion into the *At1g20990* locus is not the cause for the *fif* phenotype.

### 2.3. The fif Phenotype is Caused by a Novel Allele of LEAFY (LFY)

To identify the mutant locus genetically responsible for the *fif* phenotype, we combined a classical mapping [25,26,27] with a mapping-by-sequencing approach [28,29]. To establish a mapping population, *fif* mutant plants (No-0) were crossed in both direction with plants of the Col-0 accession. Irrespective of the crossing direction, all the F1 plant displayed a wild type phenotype. Eight F1 plants were self-fertilized and 1582 F2 plants characterized phenotypically. In accordance with the self-crossing results described above, around 25% of the F2 plants (437 of the 1582) showed the *fif* phenotype. Leaf material was harvested from 425 of the 437 F2 plants in groups of 15 to 20 individuals; in addition, leaf material from 200 F2 plants was collected individually. Genomic DNA was extracted and used for classical mapping. Using chromosome-specific INsertion and DELetion (INDEL) markers [27] the mutant locus was mapped to the q-arm of chromosome 5 (Figure 3b). Two additional INDEL markers and two single nucleotide polymorphism (SNP)-based derived cleaved amplified polymorphic sequences (dCAP) markers [25,26] limited the quantitative trait locus (QTL) responsible for the *fif* phenotype to the terminal end of chromosome 5’s q-arm (Figure 3c, dCAP S5-24: 99% No-0).

To establish the exact localization of the mutant locus, we deep-sequenced the total genome of 245 homozygous *fif* mutant plants derived from the *fif* (No-0) x WT (Col-0) crosses described above, and determined the frequencies of No-0 and Col-0 alleles along the chromosomes. Whereas the heterozygous distribution of No-0 and Col-0 sequences was found to be equal with respect to chromosomes 1 to 4 (Appendix A), there was a very significant deviation towards No-0 sequences at the terminal end of chromosome 5 (Figure 4a). A detailed examination of this 300 kb stretch revealed 100% identity with the No-0 sequence (Figure 4b). This sequence stretch conformed with the QTL identified by the classical mapping.

A detailed comparison of the *fif* and wild type No-0 sequence in this 300 kb stretch revealed a single SNP, which did not result in a silent mutation but caused a change in a codon. This SNP was also found in all the 143 individually tested *fif* mutant plants and reflected a single guanine-to-adenine exchange in the second exon of the *LEAFY* (*LFY*) gene (*At5g61850*, Figure 4c). This mutation caused a cysteine-to-tyrosine amino acid exchange at position 263 in the DNA-binding domain of the LFY protein (Figure 4d). To prove that this point mutation causes the *fif* phenotype, we transformed the *fif* mutant (No-0) with constructs coding for wild type LFY-GFP under the control of the *Arabidopsis ubiquitin 10* (*UBI10*) promoter. The functionality of C- (and N-terminal) GFP fusions of LFY was previously shown by the genetic complementation of the *lfy-12* mutant phenotype [30]. Although expressed to a very low level, LFY-GFP was detectable inside the nucleus of cells of the transgenic *fif* mutant plants and complemented the *fif* phenotype (Figure 5).

### 2.4. LFY^FIF^ Impairs DNA-Binding Capability but Shows Wild Type Intracellular Localization and Homomerization

Having identified a new *LFY* allele to be responsible for the *fif* phenotype, we next analysed the putative consequences of the Cys263-to-Tyr exchange for LFY protein properties at molecular and cellular level.

To test a putative alteration in subcellular localization, a C-terminal GFP fusion of wild type LFY and a C-terminal RFP fusion of the mutant LFY version (LFY^FIF^) were co-expressed under the control of the *UBQ10* promoter in transiently transformed *Nicotiana benthamiana* epidermal leaf cells. As shown in Figure 6a, LFY-GFP and LFY^FIF^-RFP localised to the cytoplasm and the nucleus in a similar manner. The observed fluorescence pattern of LFY-GFP and LFY^FIF^-RFP is in accordance with the pattern previously reported for the expression of fluorophore-tagged LFY fusion proteins in tobacco epidermal leaf cells [31].

Next, we tested by in vivo FRET-FLIM whether LFY protein–protein interaction, here especially LFY homomerization [31], was altered. To do so, C-terminal GFP fusions (FRET donors) and C-terminal RFP fusions (FRET acceptors) were transiently expressed, either individually (donor only) or in combination in *N. benthamiana* epidermal leaf cells and the fluorescence lifetime of the donor fusion was measured. As shown in Figure 6b, the fluorescence lifetimes of LFY-GFP and LFY^FIF^-GFP were similar in the absence of the acceptor fusions. However, the lifetimes of LFY-GFP and LFY^FIF^-GFP decreased significantly when they were co-expressed with either LFY-RFP or LFY^FIF^-RFP demonstrating homotypic (LFY-LFY, LFY^FIF^-LFY^FIF^) and heterotypic (LFY-LFY^FIF^) homomerization *in planta* (Figure 6b). There was no significant difference in the interaction properties of the homotypic and heterotypic homomers (Figure 6b).

The Cys263-to-Tyr exchange is located in the first α-helix of the LFY DNA-binding domain (Figure 4d). We, therefore, used a quantitative DNA-protein interaction ELISA approach (qDPI-ELISA; [32]) to test whether the mutation interferes with the DNA-binding capability of LFY in vitro. We expressed N-terminally GFP-tagged full-length LFY, LFY^FIF^ and GFP, in *E. coli* independently and applied the crude extracts containing the fusion proteins or GFP, in identical amounts, based on the GFP fluorescence and Western-blotting, to ELISA plates in two dilutions (Figure 7). The plates were covered with double-stranded (ds) DNA oligonucleotides representing either the LFY-binding sequence of the *AP1* promoter (*pAP1*), a mutated *pAP1* version (*pAP1m*) that is not recognized by LFY [17], a random sequence without any similarity to the LFY binding motif (*C28M12*), or were uncovered. The DNA-binding efficiency of the proteins was recorded by determining the GFP fluorescence of the bound proteins [32]. GFP-LFY exhibited a specific binding to *pAP1* and no binding to any other oligonucleotide or to the oligonucleotide-free ELISA plate (Figure 7). In contrast, GFP-LFY^FIF^, like GFP or the *E. coli* crude extract without recombinant protein, was unable to recognize *pAP1* or any other oligonucleotide (Figure 7). To exclude the possibility that the Cys263-to-Tyr exchange may alter the DNA-binding specificity we used a quantitative DPI-ELISA based in vitro approach to screen a dsDNA oligonucleotide library representing 4096 randomized DNA hexamers [33,34] with GFP-LFY- and GFP-LFY^FIF^-containing *E. coli* extracts. With this approach it is possible to narrow down the target dsDNA site of almost any DNA-binding GFP-tagged fusion protein of interest in vitro [32]. Whereas an in vitro DNA-binding sequence was obtained for GFP-LFY (5′-GGGC-3′/3′-CCCG-5′), there was no DNA-binding of GFP-LFY^FIF^ to any oligonucleotide in the library.

## 3. Discussion

In our search for novel floral genes in *Arabidopsis thaliana* we identified the *fif* Ds transposon insertion mutant in the No-0 accession in the RIKEN RAPID collection [23,24]. *fif* mutant plants display a striking floral phenotype and inflorescence architecture, as they develop aberrant and infertile primary flowers in combination with short stems that emerge from vegetative meristems in the axillars of the bracts and carry fertile secondary flowers.

The *Ds* transposon insertion in the genome of the *fif* mutant was annotated to gene *At1g20990*. However, as demonstrated here, the *Ds* transposon insertion into the *At1g20990* locus is not the cause for the *fif* phenotype. Obviously, another mutant locus generated somewhere else in the genome, most likely during transposon movement, is responsible for the *fif* phenotype. Using combined classical and genome sequencing-based mapping approaches and complementation of the mutant phenotype by *UBI10*-driven expression of wild type LFY-GFP in the *fif* mutant, the causal mutation for the *fif* phenotype was found to be in the *LFY* gene. In contrast to LFY overexpression by the strong *35S* promoter in Nossen [35], the *UBI10*-driven expression of wild type LFY-GFP in the *fif* mutant background did not cause primary shoot termination and conversion of axillary meristems into solitary flowers. We ascribe this to the very low accumulation of LFY-GFP that can hardly be detected in the nuclei of the stably transgenic *Arabidopsis* cells.

The *fif* phenotype-causing mutation is a single G-to-A exchange in the second exon of *LFY*, creating the novel, recessive *lfy* allele. The mutation causes a Cys-to-Tyr exchange at position 263 in the LFY^FIF^ amino acid sequence. The cell biological analysis of LFY-GFP and LFY^FIF^-GFP revealed an intracellular localization in the cytoplasm and nucleus of tobacco epidermal leaf cells identical to that previously reported for LFY-GFP [31]. Thus, a mis-localisation cannot be the cause of the LFY^FIF^ malfunction. In addition, as shown by quantitative FRET-FLIM interaction studies the mutation does not interfere with the oligomerization capacity of LFY that is essential for its floral function [36]. Especially the latter result was to be expected as the domain, required for oligomerization is located at the N-terminus of LFY [36].

However, our quantitative DPI-ELISA assay demonstrated that, in contrast to LFY-GFP, LFY^FIF^-GFP lost its capacity to bind to its DNA target in vitro, as it is present, for instance, in the *AP1* promoter [17]. Furthermore, the in vitro DPI-ELISA based screen for the determination of putative alterations in binding specificity did not reveal any DNA-binding activity for LFY^FIF^-GFP, whilst a defined sequence was found for LFY-GFP. This sequence did not perfectly match the LFY consensus sequence identified by, for instance, in vivo chromatin immunoprecipitation [37]. However, this discrepancy is very likely due to the design of the dsDNA oligonucleotides (e.g., the region flanking the hexameric core) used for the library and the applied in vitro conditions [33,34] that could modify LFY-GFP’s binding characteristics.

According to the available crystal structure of the DNA-bound dimer, Cys263 is well conserved between the LFY homologs of many plant species but has never previously been reported to be crucial for DNA-binding [38]. Intriguingly, Cys263 does not contribute to the physical contact of LFY with DNA; however, the α1-helix, in which Cys263 is positioned, participates in the cooperative DNA-binding of LFY, as it facilitates the establishment and stabilization of the DNA-binding domains in the minor and major grove of DNA [38]. Therefore, the change of the relatively small Cys to the bulky, aromatic Tyr might prevent the folding of the α1-helix and thereby might restrict the cooperative binding of LFY to its target DNA.

The total failure of LFY^FIF^ to bind to DNA explains the strong floral phenotype of especially the primary flowers. LFY is one of the master regulators in the floral initiation process of *Arabidopsis* (and other plant species) and controls, together with other factors and *via* a complex regulatory network, the spatiotemporal expression of downstream genes and also of the homeotic flower genes required for flower organ formation. Although only a single amino acid exchange is affected, *fif* mirrors in principle the flower phenotype of known strong *lfy* alleles. However, of the more than 15 described *lfy* alleles [12], the six alleles that show such a strong floral phenotype produce shortened LFY polypeptides caused by either premature stop codons (*lfy-1*, *lfy-6*, *lfy-7*, *lfy-8*, *lfy-11*) or a non-sense frame shift C-terminal of Gln196 (*lfy-15*). Hence, the strong phenotype of the *fif* allele needs a different explanation: LFY^FIF^ may titrate out interaction partners by binding them and not binding to DNA. Thereby, LFY^FIF^ increases its own phenotype.

The failure of LFY^FIF^ to bind to DNA also explains the growth architecture of the *fif* mutant. It has been shown [10,12] that mutations in *lfy* can cause the emergence of axillary meristems instead of floral meristems resulting in an enhanced number of side branches. In addition, the expression of a nearly full-length LFY version with weaker in vitro DNA-binding capacity under the LFY promoter dramatically reduced in vivo transcriptional activity of [LFY_HARA(∆40)_] in the Col-0 accession. This causes the formation of shoot-like structures in positions where flowers should outgrow [39].

Taken together, our data demonstrate the general importance of Cys263 for LFY function not only in floral development but also in axillary meristem outgrowth in *Arabidopsis*.

Most intriguingly, the *fif* floral phenotype appears to be specific for the No-0 accession, as, to our knowledge, it has never been reported for the Col-0 or any other accession. However, the *fif* phenotype also becomes manifest in the Col-0 accession when the *fif* locus of No-0 is transferred to Col-0. This phenomenon might be explained by differences in the spatio-temporal transcriptional activity of the No-0 and Col-0 *LFY* loci during vegetative meristem and floral development. Therefore, the *fif* phenotype may only be visible in other accessions such as Col-0 when the No-0 locus is artificially introduced into them and drives LFY^FIF^ accumulation there.

## 4. Materials and Methods

### 4.1. Plant Material and Arabidopsis thaliana Transformation

Seeds of the homozygous *Ds* transposon insertion line 15-3794-1 and the corresponding wild-type accession (No-0) were obtained from the RIKEN *Arabidopsis* Phenome Information database (RAPID; [22]. *Agrobacterium tumefaciens*-mediated transformation of the *fif* mutant with *pUGT1-LFY* leading to LHY-GFP expression and selection of transgenic lines were carried as described previously [40].

### 4.2. Plasmid Construction

Using gene-specific primers [sense (S): 5′-caccATGGATCCTGAAGGTTTCACG-3′, antisense (A): 5′-GAAACGCAAGTCGTCGCCG-3′) the cDNA of *LFY* was amplified from pSST14 (gift Jan Lohmann, University of Heidelberg, Germany) and cloned in pENTR™/D-TOPO^®^. Site-directed mutagenesis (SDM) was performed to produce the *fif* cDNA using the following primers (S: 5′-CTGTTCCACTTGTACGAACAATaCCGTGAGTTCCTTCTTCAG-3′, A: 5′-CTGAAGAAGGAACTCACGGtATTGTTCGTACAAGTGGAACAG-3′). With Gateway™ LR Clonase™ II Enzyme mix the *LFY* cDNA was inserted into pUGT1-Dest (A. Hahn, unpublished) and pB7RWG2-Dest [41] for plant expression and into pET-Dest42GFP [32] for *E. coli* expression.

### 4.3. Classical Mapping and Mapping by Genome Sequencing

Genetic mapping was accomplished using 100 phenotypic *fif* plants collected from a F2 population derived from a cross between *fif* (No-0) and Col-0. The mapping strategy and the molecular markers used to identify the causal locus were described by Pacurar, Pacurar, Street, Bussell, Pop, Gutierrez and Bellini [27]. After mapping of the chromosome arm and next-generation sequencing (NGS, see below) the point mutation was confirmed by derived cleaved-amplified polymorphic sequence primers designed by using the dCAPS Finder 2.0 software [25]. One or two mismatches were introduced in one of the used primers to incorporate an allele-specific restriction site into the PCR product. After amplification, the PCR products were digested (enzymes from Thermo Scientific, city, country, USA) following the manufacturer’s recommendations and separated on a 4% agarose gel. All used markers are listed in Appendix A.

NGS mapping was performed using a pool of 425 phenotypic *fif* plants from the crossing described above. A pool of 40 wild-type No-0 plants was sequenced to generate a genome-wide marker list and to mine the *fif* genome for acquired mutations. Isolation of genomic DNA was performed in groups up to 20 plants using the DNeasy^®^ Plant Mini Kit (QIAGEN, Venlo, Netherlands) following the manufacturer’s recommendations. DNA concentration was determined with the use of NanoDrop ND-1000 and the whole pool composed by using 100 µg DNA of each group. Sequencing was performed at the Max Planck-Genome-Centre Cologne by a HiSeq2500 (Illumina, San Diego, CA, USA) Sequencer producing ~35.000.000 read-pairs for each pool. Short reads of both pools were respectively aligned against the Col-0 reference sequence (TAIR10) and SNPs were called using *shore* pipeline (version v0.8) with *GenomeMapper* (version v0.4.4s) with default parameters [42,43]. Genome-wide SNP markers were defined with filtering for sequencing coverage and allele frequency using *SHOREmap* (version 3.0, [29,44,45]. Sliding window-based estimation of allele frequencies of the Nos allele in the pooled F2 samples and identification of a mapping interval were performed with *SHOREmap* (version 3.0) using default parameters. Comparison of the consensus calls of both pools in the 300 kb mapping interval revealed the mutation in *LFY*.

### 4.4. Localization and FRET-FLIM Studies

The indicated constructs and p19 as gene silencing suppressor were transformed into *Agrobacterium tumefaciens* strain GV3101 and infiltrated into *Nicotiana benthamiana* leaves. The localization of the fusion proteins was performed 3 days after infiltration using 488 nm or 561 nm lasers for GFP or RFP excitation, respectively, at the SP8 laser scanning microscope (Leica Microsystems GMBH) with LAS AF and SymPhoTime software using a 63x/1.20 water immersion objective [46]. FLIM data were derived from measurements of at least 20 probes for each fusion protein combination. To excite LFY-GFP and LFY^FIF^-GFP for FLIM experiments, a 470 nm pulsed laser (LDH-P-C-470) was used, and the corresponding emission was detected with a SMD Emission SPFLIM PMT from 495 to 545 nm by time-correlated single-photon counting using a Picoharp 300 module (PicoQuant, Berlin, Germany). Each time-correlated single-photon counting histogram was reconvoluted with the corresponding instrument response function and fitted against a monoexponential decay function for donor-only samples and a biexponential decay function for the other samples to unravel the GFP fluorescence lifetime of each probe. The average GFP fluorescence lifetimes as well as the standard error values were calculated using Microsoft Excel 2013. To test for homogenity of variance Levene’s test (df = 5/140, *F* = 26.298, *p* < 0.0001) was used and statistical significance was calculated by a two-tailed, all-pair Kruskal–Wallis test followed by a Steel-Dwass post hoc correction using JMP version 12.2.0 [47].

### 4.5. qDPI-ELISA, DPI-ELISA Based Screening and Western Blotting

qDPI-ELISA was performed using *E. coli* crude extracts containing GFP-tagged LFY or LFY^FIF^, GFP alone or no fluorescent protein according to Fischer, Böser, Hirsch and Wanke [32]. The sequences of the 5′-biotinylated dsDNA oligonucleotides *AP1*, *mAP1* and *C28M12* used for the immobilization on Streptavidin-coated 384 well microtiter plate are displayed in Appendix A. Before addition to the microtiter plate, the equal content of GFP-tagged fusion protein in the crude extracts was adjusted according to the GFP fluorescence using a fluorescence reader (TECAN Safire, Männedorf, Switzerland). Western blotting with GFP antibodies was performed according to Brand, et al. [48].

The quantitative DPI-ELISA based specificity screening, using a dsDNA oligo array on a 384 well microtiter plate covering all possible 4096 hexanucleotide DNA motifs was performed as described previously [32,33,34].

## Figures and Tables

**Figure 1 plants-08-00599-f001:**
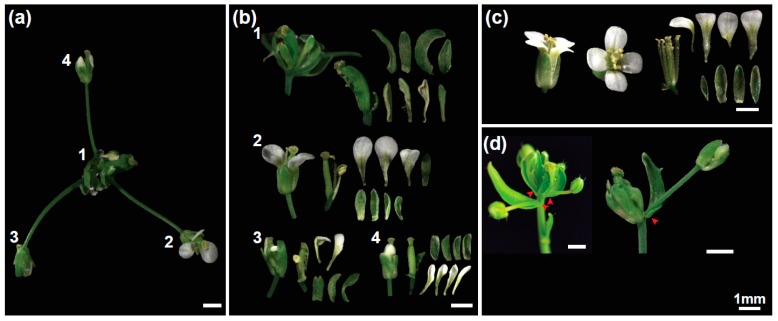
Flower phenotype of the *Arabidopsis thaliana* (No-0) *flower-in-flower* (*fif*) mutant. (**a**) Overview over a representative *fif* mutant “inflorescence” displaying different flower types 1 to 4. (**b**) Floral organs of the primary *fif* flower (1) and different secondary *fif* flowers (2–4). (**c**) Flower of the wild type No-0 accession. (**d**) Primary flowers of the *fif* mutant with stems that outgrow from axillary bract meristems (red arrow heads) and carry secondary flowers. Size bar: 1 mm.

**Figure 2 plants-08-00599-f002:**
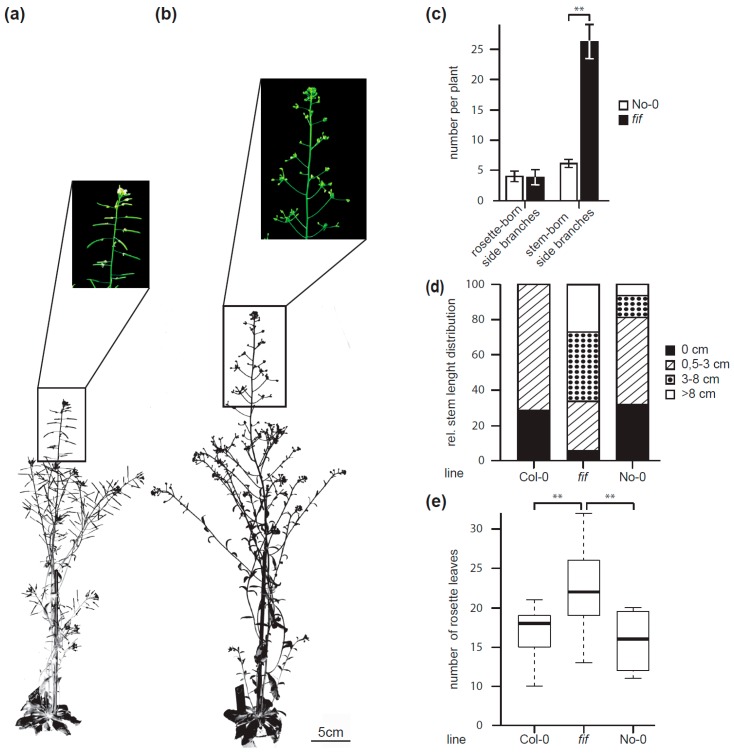
Growth habitus and flowering time of wild type No-0, wild type Col-0 and *fif* mutant plants. Overview over the growth habitus and magnification of the inflorescence of 6.5-weeks old wild type No-0 (**a**) and *fif* plants (**b**), grown side-by-side in the greenhouse. Size bar: 5.0 cm. (**c**) Number of rosette-born side branches and stem-born side branches of wild type No-0 (white bars) and *fif* (black bars) plants. Error bars indicate the standard deviation of the mean. The statistical significance (n_No-0_ = 33, n*_fif_* = 25) was testet by two-sited t-test (***: *p* = 2 × 10^−23^). (**d**) Relative distribution of stem length in wild type No-0, wild type Col-0 and *fif* mutant plants 31 days after sowing (n_Col-0_ = 7, n_No-0_ = 16, n*_fif_* = 89). (**e**) Number of rosette leaves at the onset of senescence for wild type No-0, wild type Col-0 and *fif* mutant plants. The data are presented in Box-and-Whisker plots including the median (thick line), the upper and lower quartile (+/− 25%, white boxes) and the maximum and minimum (dottet line). The statistical significance (n_No-0_ = 4, n_Col-0_ = 9, n*_fif_* = 38) was tested with ANOVA followed by a Tukey honest significant difference post-hoc test (*: *p* < 0.05; **: *p* < 0.01; ***: *p* < 0.001).

**Figure 3 plants-08-00599-f003:**
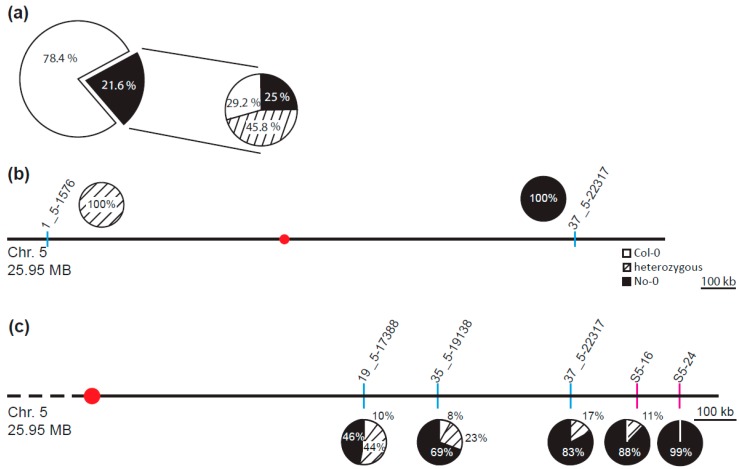
Segregation analysis and mapping of the mutant locus causal for the *fif* mutant phenotype. (**a**) Segregation of the floral phenotype and the Ds transposon insertion within the combined F_2_ population of (♀*fif* × ♂No-0) and (♀No-0 × ♂*fif*) backcrosses showing either the wild type (78.4%) or the *fif* floral phenotype (21.6%) (left) and distribution of the transposon insertions within the plants of the F_2_ population that displayed the *fif* floral phenotype (right); white circle outcut: no transposon insertion (29.2%), striped outcut: heterozygous for the Ds transposon insertion (45.8%), black outcut: homozygous for the Ds transposon insertion (25.0%). (**b**,**c**) Insertion and DELetion INDEL marker- and single nucleotide polymorphism (SNP)-based derived cleaved amplified polymorphic sequences (dCAP) marker-associated containment of the *fif* locus using a mapping population generated by a cross of the *fif* mutant (No-0) with wild type Col-0. Schematic representation of the *Arabidopsis thaliana* chromosome 5 (sizes in MB) and the localization of the chromosome-specific INDEL markers initially used for mapping (codes above blue lines) (b). Schematic representation of the q-arm of chromosome 5 and the localization of INDEL (codes above the blue lines) and SNP-based dCAP markers (codes above red lines) used for fine mapping (c). The pie charts show the distribution of the No-0 and Col-0 genotypes for chromosome 5 (b) and the q-arm of chromosome 5 (c). White circular outcut: homozygous for Col-0, striped outcut: heterozygous for Col-0/No-0, black outcut: homozygous for No-0; red dot: localization of the centromere.

**Figure 4 plants-08-00599-f004:**
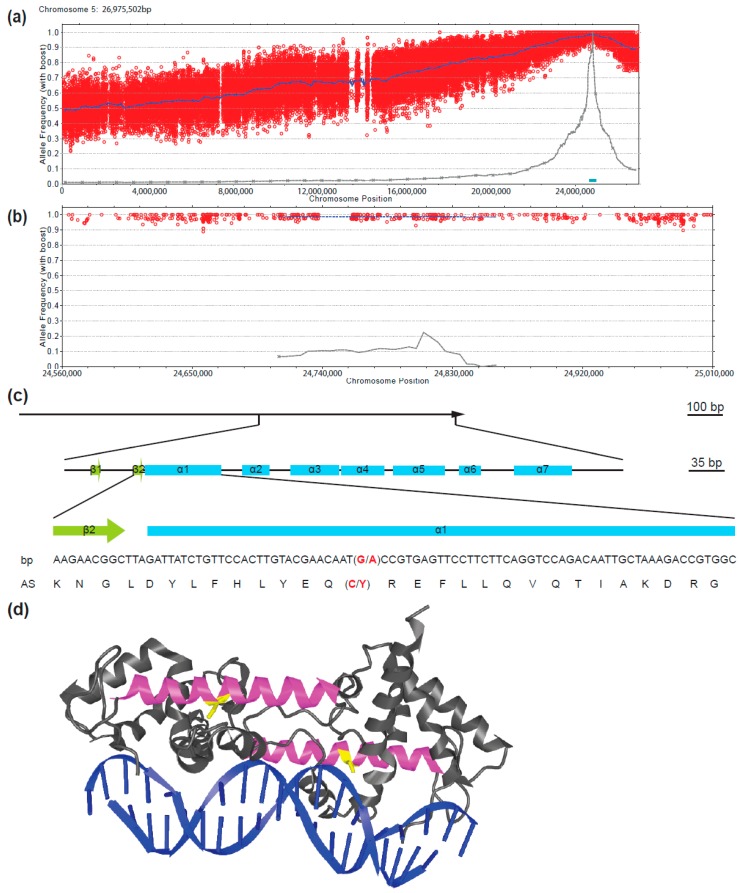
Identification of the *fif*-related SNP in the second exon of the *LEAFY* (*LFY*) locus on chromosome 5 by genome sequencing of a mapping population generated by a cross of the *fif* mutant (No-0) with wild type Col-0. (**a**) Allele frequency analysis of the No-0 genotype within chromosome 5 of the recombinant mutant pool. Each red circle refers to a SNP marker distinguishing the No-0 and Col-0 genotypes. The blue line refers to a 200 kb sliding window analysis of the allele frequencies. The brown line and blue box highlight the estimated mapping intervals (x-axis: genomic location; y-axis: Nos allele frequency). (**b**) Like (a), but only showing the 300 kb mapping interval. (**c**) Sequence of the *LFY* gene showing the *fif* SNP (G to A exchange, red) and the resulting amino acid exchange (C to Y, red) within the DNA-binding domain of the LFY protein. Green boxes: β-sheets; blue boxes: α-helices. (**d**) Structural representation of the wild type LFY dimer bound to DNA according to Hames and colleagues (2008). The α1 helix in the monomers is shown in magenta and the cysteine (C) in yellow that are mutated to tyrosine (Y) in the *fif* mutant.

**Figure 5 plants-08-00599-f005:**
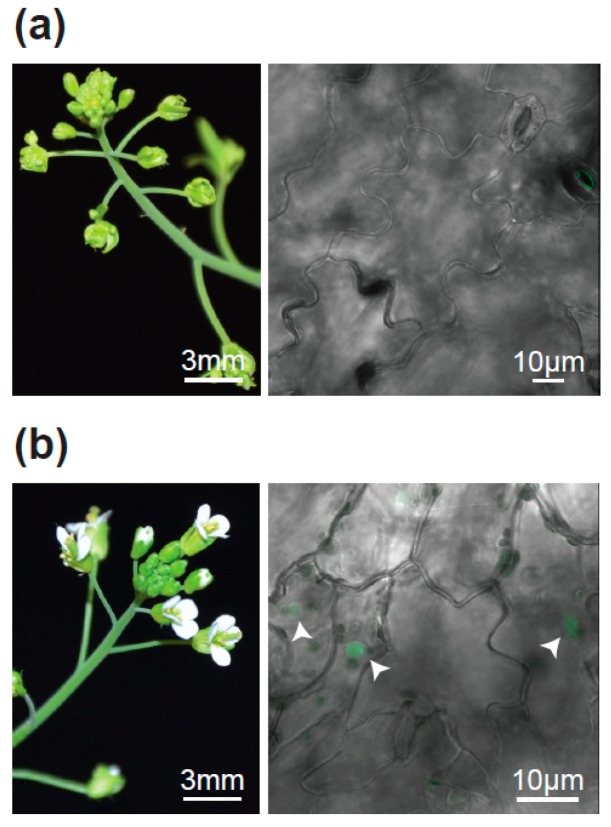
Complementation of the *fif* mutant phenotype by the *UBI10* promoter-driven expression of wild type *LFY-GFP*. (**a**) Inflorescence (left) and confocal image of epidermal cells (right) of a representative non-transformed *fif* mutant plant. (**b**) As in (a), but for a representative *LFY-GFP* transgenic plant in the *fif* mutant background. The fluorescent nuclei are highlighted by arrowheads. Three independent transgenic lines were obtained displaying nuclear localization of LFY-GFP and in parallel the complementation of the *fif* phenotype.

**Figure 6 plants-08-00599-f006:**
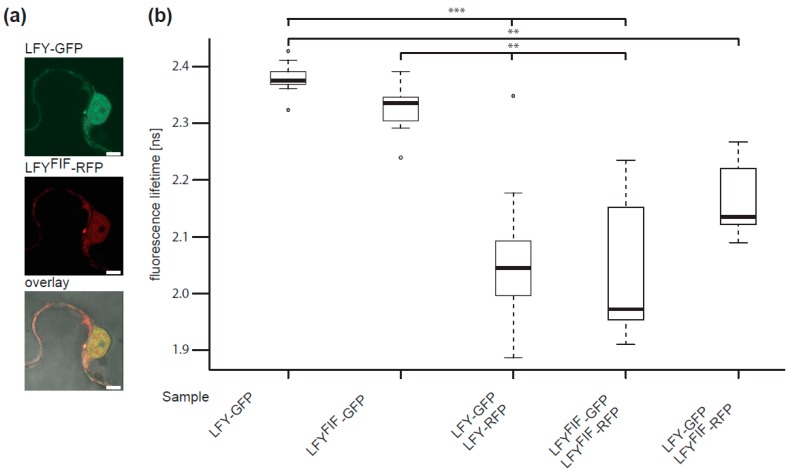
Comparative analysis of the intracellular localization and homomerization capacity of LFY and LFY^FIF^. (**a**) Confocal fluorescence images of transiently transformed *Nicotiana benthamina* epidermal leaf cells expressing LFY-GFP and LFY^FIF^-RFP in the same cell. Size bar: 5 µm. (**b**) FRET-FLIM analysis of the homo- and heterotypic interaction of LFY and LFY^FIF^. LFY-GFP or LFY^FIF^-GFP were expressed either alone or together with the indicated RFP fusions and the fluorescence lifetime of the GFP fusions measured in the nuclei. A reduction of the GFP fluorescence lifetime indicates interaction. The data are presented in Box-and-Whisker plots including the median (thick line), the upper and lower quartile (+/− 25%, white boxes), the maximum and minimum (dottet line) and outlier points (*n* > 20, each). The variance was analyzed by a Levene test and statistical significance was determined with an all-pair, two-sided Kruskal–Wallis test followed by an all-pair Steel-Dwass test (**: *p* < 0.01; ***: *p* < 0.001).

**Figure 7 plants-08-00599-f007:**
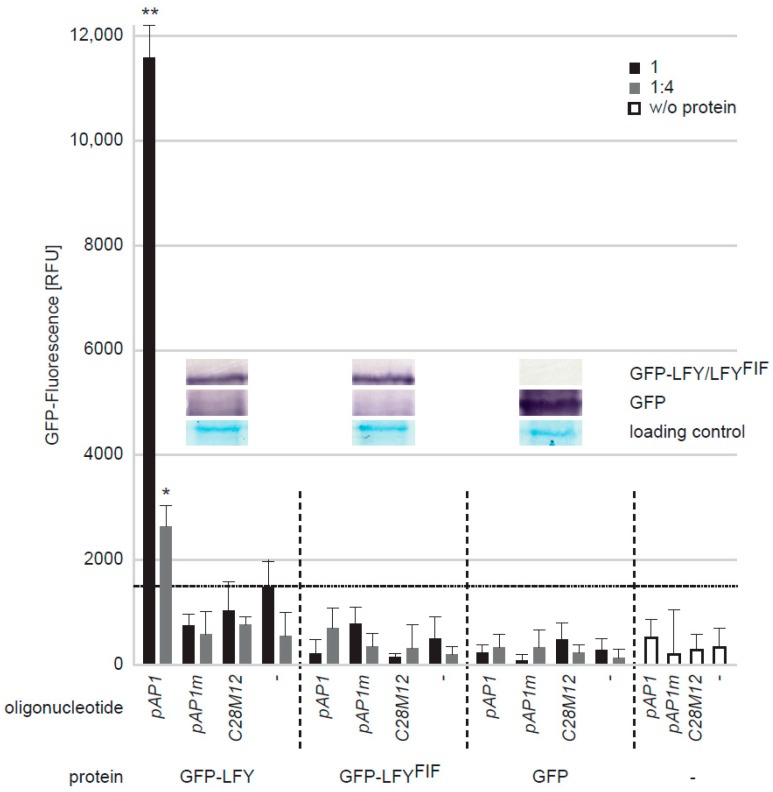
Comparative analysis of the in vitro DNA-binding capacity of LFY and LFY^FIF^ using a GFP-fluorescence-based DPI-ELISA approach. GFP-LFY, GFP-LFY^FIF^ and GFP were expressed in *E. coli*. After extraction, crude extracts containing either no recombinant protein (w/o protein) or, based on GFP fluorescence, equal amounts of GFP or the GFP fusion proteins were added to ELISA plates covered with either the double-stranded (ds) DNA oligonucleotide *pAP1*, which contains a LFY binding site, an altered version of *pAP1* (*pAP1m*), in which the binding site was mutated, a dsDNA oligonucleotide unrelated to the *pAP1* and *pAP1m* sequences (*C28M12*) or without any DNA-oligonucleotide. The amount of DNA-bound fusion protein was detected by reading out the GFP fluorescence. The crude extract was either used undiluted (black bars) or in a 1:4 dilution (grey bars). Error bars indicate the standard deviation of the mean (n = 3) and asterisk statistically significant differences to the background fluorescence (dotted horizontal line), determined by two-sided t-test (*: *p* < 0.05; **: *p* < 0.01). The inlet shows a Western-blot of the crude extracts using a GFP polyclonal antiserum for detection of GFP, GFP-LFY and GFP-LFY^FIF^ as well as a Coomassie stain as loading control.

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
