# Peer review of "The Striking Flower-in-Flower Phenotype of Arabidopsis thaliana Nossen (No-0) is Caused by a Novel LEAFY Allele"

_plants, 2019, doi:10.3390/plants8120599_

Round 1
Reviewer 1 Report
The manuscript "The striking flower-in-flower phenotype of Arabidopsis thaliana Nossen (No-0) is caused by a novel LEAFY allele", describes the characterization of a transposon insertion mutant in the Nossen accession with defects in floral development and growth. By means of classical and genome sequencing-based mapping, the mutation responsible for the fif phenotype was mapped in the LEAFY gene. A further molecular analysis of the properties of the LFYFIF protein allowed the authors to prove the molecular effect of this aminoacidic substitution.
Overall, this study is based on sound and well-presented experimental work.
There are just few minor revisions that the Authors need to address
Results
- pg.5. Comparing the sentence (lines 129/130) "an additional 49% contained the transposon insertion only heterozygously (Figure 3a)", to the legend of Figure 3a (lines 139/140) "striped outcut: heterozygous for the Ds transposon insertion (45.8 %), it seems that there is an inconsistency.
Discussion
- The sentence "The failure of LFYFIF to bind to DNA is also explains the growth architecture of the fif mutant" should be modified in "The failure of LFYFIF to bind to DNA also explains the growth architecture of the fif mutant"
Materials and Methods
- pg. 12. The first paragraph (lines 341/352) should be removed since it represents only instructions to Authors
Figures
Legend of Figure 5. LFY-GFP should be in italic
Legend of Figure S1: "The reciprocal cross (♀fif x ♂No-0) provided the identical the results." should be modified in "The reciprocal cross (♀fif x ♂No-0) provided identical results."
Author Response
Dear Reviewer,
Thank you for reviewing our manuscript and providing us with valuable hints and corrections.
Ad Results (page 5): The value of 49 % was just a typo. The value of 45.8 % is correct. We changed it accordingly.
Ad Discussion: The superfluous "is" has been removed from the sentence.
Ad Material and Methods: The first paragraph has been removed.
Ad Figure 5: LFY-GFP is in italics now
Ad Figure S1: The superfluous "the" have been removed
Reviewer 2 Report
In this manuscript a deep and very interesting characterization of a developmental mutant of Arabidopsis is reported. I find the results useful to increase the our knowledge about a key transcription factor LEAFY and therefore, I retain acceptable this report for the Journal. As further comment, the expression profile, in the mutant, of important genes encoding relevant transcription factors in the control of plant form such as Lateral Suppressor, could be also interesting for the future readers. Anyway, in my, opinion, the data presented in the present manuscript are sufficient to consider the experimental work good and valid.
Author Response
Dear Reviewer,
Thank you for reviewing our manuscript and the helpful comments. Some expression data of LFY target gene would be desirable, but we wanted to focus on the identification genetic cause of the fif phenotype and the molecular background of how the mutant LFY protein mediates the phenotype (altered DNA binding activity). We hope for your understanding.
Sincerely yours,
Klaus Harter